# A Simple Romance between Multi-exit Vision Transformer and Token Reduction

**Dongyang Liu**[1,2]**, Meina Kan**[1,2]**, Shiguang Shan**[1,2,3]**, Xilin CHEN**[1,2]
[1] Key Lab of Intell. Info. Process., Inst. of Comput. Tech., CAS
[2] University of Chinese Academy of Sciences  [3] Peng Cheng Laboratory
{liudongyang21s, kanmeina, sgshan, xlchen}@ict.ac.cn

## Abstract

Vision Transformers (ViTs) are now flourishing in the computer vision area. Despite the remarkable success, ViTs suffer from high computational costs, which greatly hinder their practical usage. Token reduction, which identifies and discards unimportant tokens during forward propagation, has then been proposed to make ViTs more efficient. For token reduction methodologies, a scoring metric is essential to distinguish between important and unimportant tokens. The attention score from the [CLS] token, which takes the responsibility to aggregate useful information and form the final output, has been established by prior works as an advantageous choice. Nevertheless, whereas the task pressure is applied at the end of the whole model, token reduction generally starts from very early blocks. Given the long distance in between, in the early blocks, [CLS] token lacks the impetus to gather task-relevant information, causing somewhat arbitrary attention allocation. This phenomenon, in turn, degrades the reliability of token scoring and substantially compromises the effectiveness of token reduction. Inspired by advances in the domain of dynamic neural networks, in this paper, we introduce Multi-Exit Token Reduction (METR), a simple romance between multi-exit architecture and token reduction—two areas previously considered orthogonal. By injecting early task pressure via multi-exit loss, the [CLS] token is spurred to collect task-related information in even early blocks, thus bolstering the credibility of [CLS] attention as a token-scoring metric. Additionally, we employ self-distillation to further refine the quality of early supervision. Extensive experiments substantiate both the existence and effectiveness of the newfound chemistry. Comparative assessments also indicate that METR outperforms state-of-the-art token reduction methods on standard benchmarks, especially under aggressive reduction ratios.

## 1 Introduction

The transformer architecture was first introduced in (Vaswani et al., 2017) for natural language processing. Due to advantages like flexible input format and outstanding scaling performance (Zhai et al., 2022; Kaplan et al., 2020), Transformer has now become a universal architecture widely adopted by diverse deep learning areas (Carion et al., 2020; Radford et al., 2021; Alayrac et al., 2022; Radford et al., 2023). In computer vision, starting from the monumental work Dosovitskiy et al. (2020), transformers have also achieved state-of-the-art performance on a variety of tasks.

Despite the remarkable success, vision transformers suffer from high computational cost, which greatly hinders their practical usage, especially under resource-constrained circumstances. In pursuit of more efficient ViTs, some works revisit the traditional model compression techniques like distillation (Yang et al., 2022; Ren et al., 2023; Liu et al., 2022), pruning (Yang et al., 2021), quantization (Lin et al., 2021), etc, and some others turn to efficient architecture design, bringing up innovations like pyramid architecture (Wang et al., 2021) and sliding window attention (Liu et al., 2021). Among these ViT compression efforts, token reduction, which enjoys special affinity with the characteristics of the transformer architecture and the sparse nature of visual information, has emerged as a promising approach. Due to the quadratic complexity of self-attention operation w.r.t. token number, reducing the number of tokens can greatly alleviate the computation burden of ViTs.

| Block 4 | Block 7 | Block 10 | Block 4 | Block 7 | Block 10 |

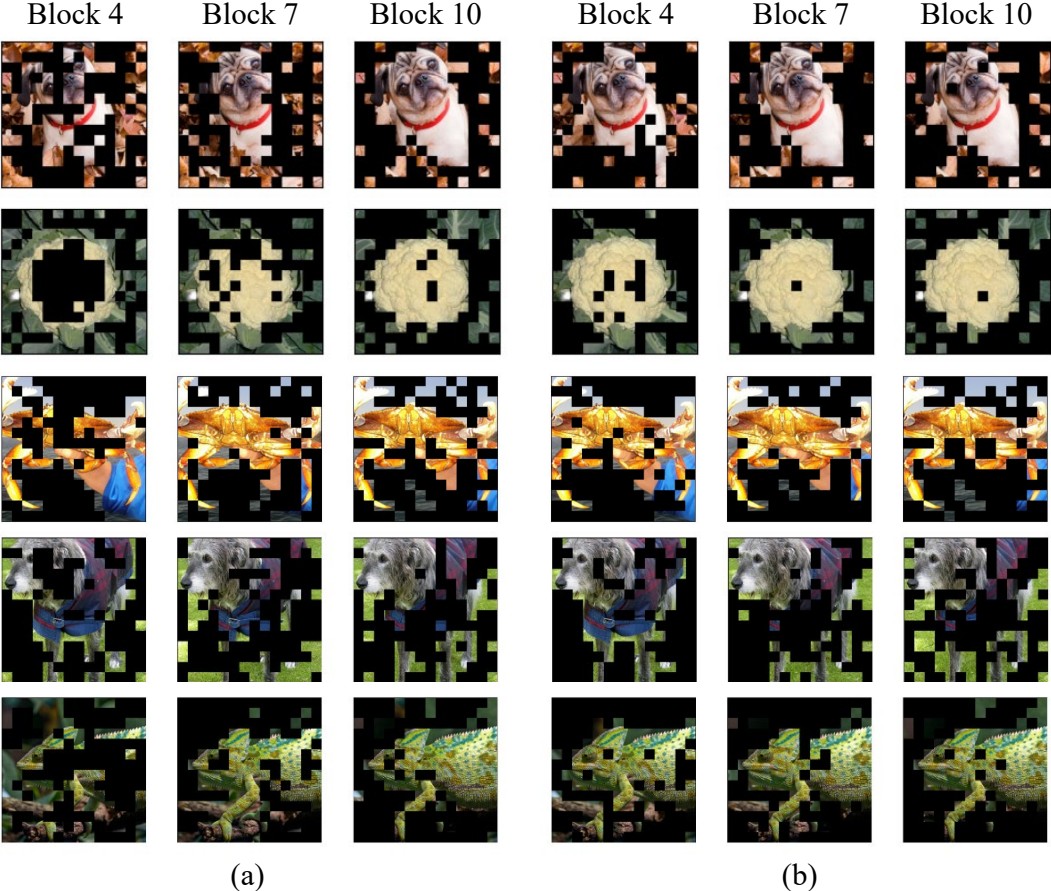

(a)                              (b)

Figure 1: Visualization of [CLS] attention. We use DeiT-S and investigate the 4-*th*, 7-*th*, and 10-*th* blocks. Token with top 50% [CLS] attention scores are kept, while others are masked. (a) Official DeiT-S model (b) Further fine-tune the official model for 30 epochs with multi-exit loss.

The methodology design of token reduction can be roughly decomposed into three sub-problems:
① How to score the importance of each token, so that the least important ones can be pruned?
② How is token removal implemented exactly (directly discard, merge into reserved ones, etc.)?
③ How to allocate the number of reduced tokens in each block?
Among these points, the first one plays a pivotal role. After a period of exploration from the research community, the attention score from the [CLS] token has been proved to be the superior choice (Liang et al., 2021; Chen et al., 2023b; Xu et al., 2022; Haurum et al., 2023). Since the [CLS] token bears the responsibility for forming the final result, it has to gather important task-relevant information from the patch tokens. Therefore, how much the [CLS] token attends to a patch token naturally becomes a reasonable proxy for the importance of the patch token.

Nevertheless, whereas the task pressure is applied at the end of the whole network, the token reduction operation needs to commence from very shallow blocks to achieve significant acceleration. Therefore, while in the last few blocks, the task pressure imposes the [CLS] token a strong impetus to pay more attention to the important informative tokens, it is doubtful if, within the blocks far before, the [CLS] token still has the abundant motivation to prioritize the most important tokens. For example, the [CLS] token may otherwise defer the summarization of important information to the last few blocks, leaving the allocation of attention score in early blocks somewhat arbitrary. We empirically validate that this is indeed the fact: as shown in Fig. 1(a), tokens with top 50% [CLS] attention scores in block 10 are mostly those containing informative cues such as the head part of the dog, but in block 4, many of these informative tokens are assigned with low attention scores (within the bottom 50%). For normal neural networks, this is okay. However, for token reduction methods, such a phenomenon significantly degrades the reliability of token scoring and substantially compromises the effectiveness of token reduction methods.

Is it possible to make [CLS] attention in early blocks as indicative of token importance as in the last block? In this paper, we reveal that the multi-exit architecture, a technique typically adopted in the domain of dynamic neural network (Han et al., 2021b), is a good solution to achieve this goal. Both multi-exit architecture and vision transformer token reduction are established topics and have been investigated independently in their respective domains. *However, we find that there exists some unexplored special chemistry between these two topics*. We thus propose Multi-Exit Token Reduction (METR), an extraordinarily simple yet highly effective and efficient combination. With the multi-exit architecture, pressure from the target task permeates through to the [CLS] token in even earlier transformer blocks, forcing it to collate useful information from patch tokens immediately. The attention score thus makes a better reflection of token significance, ensuring accurate token reduction. Additionally, we adopt self-distillation to further refine the quality of early supervision. Note that while our method aims to improve the sub-problem ① of token reduction, we empirically show that METR is compatible with recent progress targeting the other two sub-problems (②③).

Overall, *our main contributions are summarized as below*:

1. We identify the inconsistency between [CLS] attention and token significance in early ViT block, which greatly harms the performance of token reduction methods.

2. We propose METR, a simple romance between multi-exit vision transformer and token reduction, to alleviate the aforementioned inconsistency, and further employ self-distillation to improve the quality of early supervision.

3. Extensive experiments substantiate both the existence and effectiveness of the newfound chesmistry between mutli-exit and token reduction. Comparative assessments also indicate that METR outperforms state-of-the-art token reduction methods, especially under high reduction ratio.

## 2  RELATED WORK

### 2.1  VISION TRANSFORMER COMPRESSION

In the realm of computer vision, convolutional neural networks (CNNs) once held sway (Simonyan & Zisserman, 2014; He et al., 2016; Sandler et al., 2018); however, the advent of the Vision Transformer (ViT) family (Dosovitskiy et al., 2020; Touvron et al., 2021; Yuan et al., 2021; Han et al., 2021a) has heralded a new era. The model's unparalleled flexibility, versatility, and performance have driven an increasing number of researchers to employ vision transformers for a diverse range of computer vision tasks (Carion et al., 2020; Radford et al., 2021; Li et al., 2022).

However, like other deep neural networks, vision transformer suffers from the conundrum between model performance and model complexity. Model compression techniques are then investigated to fight for a better balance. Among the explorations, some works revisit the architecture-agnostic approaches like distillation (Yang et al., 2022; Ren et al., 2023), pruning (Yang et al., 2021), quantization (Lin et al., 2021), and neural architecture search (Chen et al., 2021), making them readily applicable to the ViT family. Conversely, the distinctive attributes of the transformer architecture *per se* provide unique opportunities for model compression, among which token reduction has emerged as a particularly promising avenue for research.

### 2.2  TOKEN REDUCTION

The technique of token reduction relies on two favorable properties of the vision transformer. Firstly, transformers can deal with mutable sequence length and patch-level sparsity leads to real acceleration without the need of special hardware / algorithm support (Note that this advantage has also been exploited by works like MAE (He et al., 2022)). Secondly, the information in the image is unevenly and sparsely distributed among patches, making token reduction theoretically feasible.

The core problem of token reduction is how to score the significance of each token so that we can decide which to discard and which to retain. Prior works can be roughly divided into the parametric and the non-parametric genres. Parametric scoring strategies introduce specialized policy networks to evaluate the significance of each token. For example, DynamicViT (Rao et al., 2021), SPViT (Kong et al., 2021), and IA-RED$^2$ (Pan et al., 2021) insert additional token importance prediction modules between transformer blocks, AViT (Yin et al., 2022) implicitly merges the policy network into the

original model, making one feature dimension specially responsible for representing token importance. AdaViT (Meng et al., 2022) generalizes the aforementioned idea, incorporating token-level, head-level, and block-level sparsity into a unified framework. Since discarding or retaining a token is inherently a discrete problem, re-parameterization tricks (Jang et al., 2016) or reinforcement learning (Sutton & Barto, 2018) are indispensable for optimization. However, the former hampers training-time acceleration because tokens can only be masked rather than really discarded, while the latter incurs complicated training curriculum and unstable performance.

On the other hand, non-parametric methods rely on off-the-shelf indicators to reflect token importance. Since there are no parameters to update within the scoring function, training-time acceleration is generally accessible. The attention score from the [CLS] token or its slight variance is the predominant choice of this genre (Liang et al., 2021; Yin et al., 2022). Prior works have validated its superiority (Haurum et al., 2023), and multiple works have based themselves on this metric, making advances from other aspects for creating new SOTA (Chen et al., 2023b; Fayyaz et al., 2022; Xu et al., 2022; Chen et al., 2023a). However, the reliability of [CLS] attention itself and ways for enhancing it, which is the focus of this paper, has been rarely investigated. In this paper, we point out the inconsistency between [CLS] attention and real token significance, and then propose a methodology to fix it. Meanwhile, the efforts made in this paper are orthogonal to recent advances achieved in other aspects of the token reduction problem and can be seamlessly combined together.

### 2.3 MULTI-EXIT NETWORK

Adding early-exit heads to neural networks is a long-existing topic in deep learning for various purposes. As early as the onset of the current deep learning wave, multi-exit architectures were employed to tackle the vanishing gradient problem in deep neural networks (Wang et al., 2015). At present, one typical purpose is to make neural network dynamic (Teerapittayanon et al., 2016; Bolukbasi et al., 2017; Han et al., 2021b), so that for easy samples, whose early results are already accurate, the remaining blocks can be skipped for higher efficiency. With the boom of transformer in the vision community, Bakhtiarnia et al. (2021) also applied the idea of multi-exit dynamic network to vision transformer. Different from existing works, in this paper, we open up a novel utility of the multi-exit structure, revealing its special chemistry with transformer token reduction.

## 3 METHOD

In this section, we first briefly review the architecture of vision transformer, focusing on components related to our method. We then introduce how to inject early task pressure to [CLS] tokens in early transformer blocks, making [CLS] attention a more accurate token importance indicator. Finally, we show how to use self-distillation to costlessly enhance the final performance.

### 3.1 VISION TRANSFORMER OVERVIEW

Given an input image, the ViT architecture first divides it into patches and then projects the patches into token embeddings. An extra [CLS] token is then added to the beginning of the token sequence, forming $X_0$, the input to the stack of transformer blocks:

$$\boldsymbol{X}_L = \text{Block}_L \cdot \text{Block}_{L-1} \cdots \text{Block}_2 \cdot \text{Block}_1(\boldsymbol{X}_0), \quad \boldsymbol{X}_0 = [\boldsymbol{x}_0^c, \boldsymbol{x}_0^1, \boldsymbol{x}_0^2, \cdots, \boldsymbol{x}_0^n]^\top,$$

where $\text{Block}$ means a transformer block, $L$ is the number of blocks, $n$ is the number of image patches, and superscript $c$ means the [CLS] token. $\boldsymbol{X} \in \mathbb{R}^{[n+1,d]}$ comprises of the token features $\boldsymbol{x} \in \mathbb{R}^d$ for one [CLS] token and $n$ patch tokens, and each $\boldsymbol{x}$ contains $d$ elements. The transformer block typically consists of Norm layer, Multi-Head Self-Attention (MHSA) layer, and Feed-Forward Network (FFN) layer. Considering the $i$-th block ($\text{Block}_i$), the computation graph therein is:

$$\boldsymbol{X}_{i\text{-}attn} = \boldsymbol{X}_i + \text{MHSA}(\text{NORM}(\boldsymbol{X}_i)), \tag{1}$$
$$\boldsymbol{X}_{i+1} = \boldsymbol{X}_{i\text{-}attn} + \text{FFN}(\text{NORM}(\boldsymbol{X}_{i\text{-}attn})). \tag{2}$$

In the MHSA layer, every token serves as query to attend to other tokens, gathering information by absorbing value vectors with weights defined by query-key product. Considering the [CLS] token as query, the corresponding [CLS] attention $\boldsymbol{A}^c \in \mathbb{R}^n$ is computed as follows:

$$\boldsymbol{A}^c = \text{Softmax}\left(\boldsymbol{q}^c \boldsymbol{K}^\top / \sqrt{d'}\right), \tag{3}$$

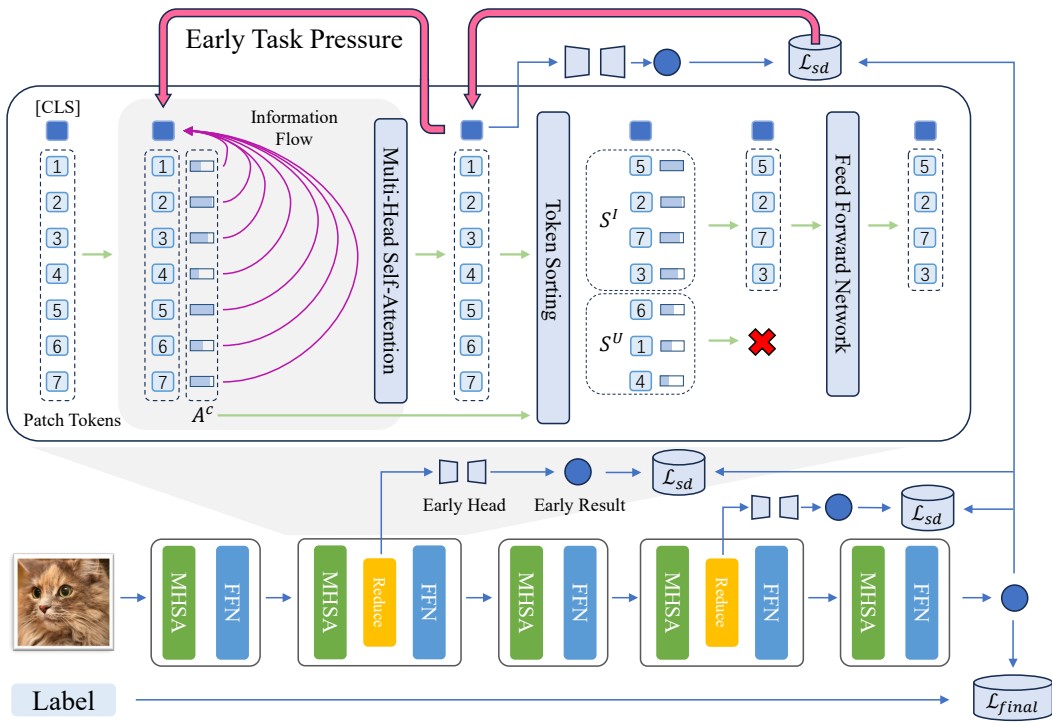

Figure 2: An overview of METR. In the attention layer, information from patch tokens flows into the [CLS] token (purple arrow), with the [CLS] token's attention score, $\boldsymbol{A}^c$, defining the contributing ratio. In each transformer block where token reduction is applied, $\boldsymbol{A}^c$ separates the input tokens into the important set $S^I$ and the unimportant set $S^U$; token reduction is then applied following the principle of retaining $S^I$ while discarding $S^U$. To ensure that $\boldsymbol{A}^c$ accurately reflects the importance of each token for the task at hand, we introduce the self-distillation loss $\mathcal{L}_{sd}$ for injecting early task pressure, which spurs the [CLS] token to turn to informative tokens for forming the correct answer.

where $d'$ is the dimension of an attention head, $\boldsymbol{q}^c \in \mathbb{R}^{d'}$ is the query vector of the [CLS] token, and $\boldsymbol{K} \in \mathbb{R}^{[n,d']}$ denotes the key matrix. With slight abuse of notation, here we make $\boldsymbol{K}$ consist of only patch tokens (i.e. [CLS] excluded).

Finally, at the end of the whole vision transformer, $\boldsymbol{x}_L^c$, the feature in $\boldsymbol{X}_L$ corresponding to the [CLS] token, is picked out and fed to the final classification head for task result.

$$\boldsymbol{O}_{final} = H_{final}(\boldsymbol{x}_L^c), \tag{4}$$

$$\boldsymbol{L}_{final} = \text{CE}(\boldsymbol{O}_{final}, y), \tag{5}$$

where $H_{final}$ is the final classification head, CE is cross-entropy loss, and $y$ is ground-truth label.

## 3.2 MULTI-EXIT TOKEN REDUCTION

For a given block, suppose there are $n$ input tokens and we want to reduce the number to $m$, we first propagate through the MHSA layer following Eq. 1, obtaining the intermediate feature $\boldsymbol{X}_{attn}$ and the [CLS] attention $\boldsymbol{A}^c$ (we omit the block index $i$ for notation convenience), and then use $\boldsymbol{A}^c$ as the metric to divide the $n$ input tokens to $m$ important tokens and $(n-m)$ unimportant tokens.

$$S^I = \{x_{attn}^p \in \boldsymbol{X}_{attn} | a^{c,p} \in \text{top}(\boldsymbol{A}^c, m)\}, \quad S^U = \boldsymbol{X}_{attn} - S^I. \tag{6}$$

$S^I$ and $S^U$ denote the set of important and unimportant tokens, respectively. $p$ indexes a patch token, $a^{c,p}$ is its corresponding attention score in $\boldsymbol{A}^c$, and $\boldsymbol{x}_{attn}^p$ is the corresponding token feature in $\boldsymbol{X}_{attn}$. The $\text{top}(\cdot, m)$ operator selects the largest $m$ elements from $(\cdot)$. The action of token reduction then follows the general principle to retain the tokens in $S^I$, while discarding those in $S^U$.

The quality of token reduction is hence dominated by the consistency between [CLS] attention ($\boldsymbol{A}^c$) and the real significance of each token. However, the extent of consistency, as shown in Fig. 1(a),

Table 1: *Off-the-shelf* token reduction performance on ImageNet. While the models have not undergone any token-reduction aware training, they are tested directly with token reduction applied. Throughput is evaluated on one RTX 4090 with batch size 512.

| Keep Ratio | 1.0 | 0.7 | 0.6 | 0.5 | 0.4 | 0.3 |
|---|---|---|---|---|---|---|
| GFLOPs | 4.6 | 3.0 | 2.6 | 2.3 | 2.0 | 1.8 |
| Throughput (imgs/s) | 2360 | 3582 | 4158 | 4712 | 5256 | 5864 |
| DeiT-S | 79.80 | 78.52 | 76.89 | 74.10 | 68.67 | 57.88 |
| DeiT-S (Multi-Exit) | 79.78 | 78.76 | 77.55 | 75.12 | 70.76 | 61.19 |
| $\Delta$ | -0.02 | +0.24 | +0.66 | +1.02 | +2.09 | +3.31 |

is in fact unsatisfactory. Specifically, while within the deeper blocks, most of the important tokens are given relatively high [CLS] attention score, the consistency degrades significantly when the block goes shallower. We attribute this phenomenon to the lack of task pressure in the early blocks: given the refined patch features in the last several blocks, it may be enough for the [CLS] token to collect task-relevant information majorly in these blocks, leaving [CLS] attention in shallow blocks relatively arbitrary. For token-reduction transformers, this becomes a fatal drawback.

In order to calibrate [CLS] attention, we need to make the pressure penetrate through to the early transformer blocks, so that the [CLS] token *always* bears the urgent impetus to collect as much task-relevant information and it could. Specifically, we make the transformer network multi-exit, adding an early-exit head $H_i$ within every block $i$ where token reduction is employed. As shown in Fig. 2, the early-exit head takes $\boldsymbol{x}^c_{i\text{-}attn}$, the [CLS] token in $\boldsymbol{X}_{i\text{-}attn}$, as input, comprising of a two-layer bottleneck MLP, outputting an intermediate result $\boldsymbol{O}_i$:

$$\boldsymbol{O}_i = H_i(\boldsymbol{x}^c_{i\text{-}attn}), \tag{7}$$

early task pressure is then applied by the multi-exit loss $\mathcal{L}_{me}$:

$$\mathcal{L}_{me} = \frac{1}{|\mathcal{B}_r|} \sum\nolimits_{i \in \mathcal{B}_r} \text{CE}(\boldsymbol{O}_i, y), \tag{8}$$

where $\mathcal{B}_r$ denotes the set of blocks where token reduction is involved, and $|\mathcal{B}_r|$ is its cardinality. In this way, in every block where token reduction is employed, the [CLS] token is compelled to provide a result given existing patch features immediately. The [CLS] attention thus becomes a better reflection of token significance, as shown in Fig. 1(b).

### 3.3 SELF-DISTILLATION

While the multi-exit loss in Eq. 8 can already effectively improve the reliability of early [CLS] attention, the high-quality final answer, $\boldsymbol{O}_{final}$, offers an opportunity for yet another free lunch. To further facilitate the efficiency and effectiveness of METR, we propose the self-distillation loss $\mathcal{L}_{sd}$ to enhance the quality of supervision for early-exit heads.

$$\mathcal{L}_{sd} = \frac{1}{|\mathcal{B}_r|} \sum\nolimits_{i \in \mathcal{B}_r} \text{KL}\left(\boldsymbol{O}_i, \text{detach}\left[\boldsymbol{O}_{final}\right]\right), \tag{9}$$

where $\text{detach}[\cdot]$ denotes the stop gradient operation. Here for simplicity, we use the KL-Divergence based KD loss (Hinton et al., 2015) for distillation, but its advanced variants (Zhao et al., 2022) may also be applicable. Finally, the total loss for training / fine-tuning a token-reduction transformer with METR becomes:

$$\mathcal{L}_{total} = \mathcal{L}_{final} + \alpha \mathcal{L}_{sd}. \tag{10}$$

Note that after training, all of the early-exit heads $H$ are removed. Therefore, no extra inference cost is introduced by METR.

## 4 EXPERIMENT

In this section, we first validate the existence of the claimed special chemistry between multi-exit architecture and token reduction in Sec. 4.1. We then delve into the design choices of METR and

Table 2: Performance on ImageNet after 30 epochs of token-reduction fine-tuning.

| Keep Ratio | 0.7 | 0.6 | 0.5 | 0.4 | 0.3 |
|---|---|---|---|---|---|
| GFLOPs | 3.0 | 2.6 | 2.3 | 2.0 | 1.8 |
| DeiT-S | 79.32 | 79.06 | 78.47 | 77.68 | 76.34 |
| DeiT-S (Multi-exit) | 79.49 | 79.22 | 78.72 | 77.91 | 76.72 |
| $\Delta$ | +0.17 | +0.16 | +0.25 | +0.23 | +0.38 |

analyze its performance in different scenarios in Sec. 4.2. Finally, we compare METR with existing token reduction methods to prove its effectiveness in Sec. 4.3. Experiemnts are conducted on the ImageNet (Deng et al., 2009) dataset using DeiT (Touvron et al., 2021) and MAE (He et al., 2022) models. Unless otherwise specified, we base METR on EViT (Liang et al., 2021), reduce the number of tokens according to the specified *keep ratio* (e.g. 30% of the tokens are reduced when keep ratio set to 0.7) in the 4-*th*, 7-*th*, and 10-*th* blocks, and fuse the unimportant tokens into one additional token. Similarly, for other experiments where we base METR on DiffRate (Chen et al., 2023b), we inherit their allocation of token number and their exact token reduction action. See appendix for detailed experiment settings.

## 4.1 EXSITENCE OF THE CHESMISTRY

### 4.1.1 OFF-THE-SHELF TOKEN REDUCTION ON MULTI-EXIT MODEL

We first fine-tune the officially pre-trained DeiT-S model with $\mathcal{L}_{total}$ in Eq. 10 for 30 epochs, *without incorporating any token reduction operations*. Subsequently, we directly add the token reduction operation to the model at inference time. Since the model has not undergone any token-reduction aware training, this forms an *off-the-shelf* evaluation setting. Based on this setting, we compare the performance of the DeiT-S model before and after muti-exit finetuning in Tab 1.

After 30 epochs of multi-exit tuning, the normal accuracy (i.e. without token reduction) of the model is almost unchanged (79.80% v.s. 79.78%). *However, the model's robustness to token reduction improves discernibly*, manifesting an average increase of 1.46%, and this margin of improvement amplifies with the application of more aggressive token reduction strategies. This empirical evidence lends credence to our argument that the incorporation of early task pressures augments the alignment between the [CLS] attention score and the token significance.

### 4.1.2 VISUALIZATION

To elucidate how the model's robustness to token reduction has been augmented through 30 epochs of multi-exit fine-tuning, we present visualizations of [CLS] attention maps, revealing insights into the mechanism at work. As depicted in Fig 1(a), for the original DeiT-S model, the [CLS] attention pattern possesses a relatively high correlation with the informativeness of individual image patches in later blocks, such as block 10. However, *this correlation substantially diminishes in earlier blocks*, for example, block 4. Conversely, after 30 epochs of multi-exit tuning (Fig. 1(b)), the congruence between attention patterns and token significance is notably strengthened, particularly in the shallower blocks. Consequently, the [CLS] attention in the fine-tuned model serves as a more reliable metric for token scoring, resulting in less accuracy drop after token reduction.

### 4.1.3 FURTHER FINE-TUNE MULTI-EXIT MODEL WITH TOKEN REDUCTION

Token-reduction models typically require an additional fine-tuning phase with token reduction explicitly incorporated to fully realize their potential. To ascertain whether the multi-exit model obtained in Sec. 4.1.1 has, in a broader context, become more amenable to token reduction, we compare it with the original DeiT-S model within this fine-tuning setting. This time, both models undergo an additional 30 epochs of fine-tuning where token reduction is explicitly employed. Early exit losses are excluded from this phase to maintain a fair comparison between the two models. As shown in Tab. 2, while multi-exit loss no longer exists in this stage of fine-tuning, its influence lasts and leads to better performance. The stable advantage of the multi-exit model supports the special chemistry between multi-exit architecture and token reduction.

Table 3: Ablation study over the design choices of METR. We start from pre-trained DeiT-S model and fine-tune for 30 or 100 epochs. EE means early exit and SD means self-distillation. Top-1 accuracy (%) on ImageNet validation set is reported.

| Epochs | Setting | Keep Ratio | | | | |
|---|---|---|---|---|---|---|
| | | 0.7 | 0.6 | 0.5 | 0.4 | 0.3 |
| 30 | ① EViT | 79.32 | 79.06 | 78.47 | 77.68 | 76.34 |
| | ② + EE | 79.51 | 79.28 | 78.82 | 78.08 | 76.79 |
| | Δ | 0.19 | 0.22 | 0.35 | 0.40 | 0.45 |
| | ③ + EE & SD | 79.64 | 79.45 | 78.94 | 78.23 | 76.90 |
| | Δ | 0.32 | 0.39 | 0.47 | 0.55 | 0.56 |
| 100 | ④ EViT | 79.58 | 79.23 | 78.85 | 78.17 | 77.13 |
| | ⑤ + EE & SD | 80.17 | 79.86 | 79.62 | 78.96 | 78.02 |
| | Δ | 0.59 | 0.66 | 0.77 | 0.79 | 0.89 |

## 4.2 ABLATION STUDY

In Sec. 4.1, we split up the multi-exit and the token-reduction fine-tuning process for clearer conclusion. In practice, however, such a split design is inefficient and unnecessary. **For all remaining experiments in this paper, we merge the above two fine-tuning stages into a single stage**. Based on this ground, in this section we ablate over some of the design choices.

### 4.2.1 ONE-STAGE TOKEN-REDUCTION FINE-TUNING WITH MULTI-EXIT LOSS

As shown in Tab. 3 (index ① and ②), by introducing early task pressure into the token-reduction fine-tuning process, the performance is considerably enhanced with no additional inference cost and negligible additional training cost (for early heads) imposed. Again, *the advantage of ② becomes increasingly large as the pruning becomes more and more aggressive*. This is intuitive: to prevent significant information loss under aggressive pruning, a stringent token scoring metric is necessary, so that the most informative tokens can be retained.

### 4.2.2 SELF-DISTILLATION

Contrary to hard-labeled-based $\mathcal{L}_{me}$, we instead use the self-distillation loss $\mathcal{L}_{sd}$ for early task pressure. Such a substitution incurs no additional computation cost, and we empirically find that there's even no need to modify the hyper-parameter $\alpha$. After the replacement. As shown in Tab. 3 (index ③), METR outperforms EViT by 0.46% on average in this setting.

### 4.2.3 LONGER FINE-TUNING

Since 30 epochs may not be enough to fully optimize the multi-exit task, we further investigate the performance of METR with longer (100 epochs) fine-tuning. See ④ and ⑤ in Tab. 3 for results. As shown in the table, *the advantage of METR is magnified with the application of longer fine-tuning recipe*, and the average improvement reaches 0.74%.

### 4.2.4 NUMBER OF MULTI-EXIT HEADS

Considering that methods like DiffRate (Chen et al., 2023b) conduct token reduction in 10 out of 12 transformer blocks, is it optimal to allocate 10 early heads, one for each of them? Interestingly, the answer is *yes*. As shown in Tab 4, on the one hand, an exit head not only influences the block it's in, but also influences the preceding blocks, so more early heads help. On the other hand, such influence is weaker than directly adding early exit heads in these blocks *per se*.

Table 4: Effect of early head number on METR performance. Model: ViT-B with GFLOPs = 8.7. We base METR on DiffRate.

| Early Heads | DeiT | MAE |
|---|---|---|
| All | 81.11 | 82.42 |
| Every 2 | 80.94 | 82.34 |
| Every 3 | 80.81 | 82.29 |

Table 5: Comparison with token-reduction methods on ImageNet (Deng et al., 2009). We report our reproduced accuracy for EViT and DiffRate, and the author-reported accuracy for others. ‡ means fusing token reduction into model training (i.e. training from scratch for 300 epochs for DeiT, and training from self-supervised checkpoint for 100 epochs for MAE), † means starting from supervisely trained checkpoints and further fine-tune for 100 epochs, and for those without special notation we start from supervised checkpoints and fine-tune for 30 epochs.

| Model | Method | GFLOPs | Acc. | Model | Method | GFLOPs | Acc. |
|-------|--------|--------|------|-------|--------|--------|------|
| ViT-S (DeiT) | *Baseline* | *4.6* | *79.82* | ViT-B (DeiT) | *Baseline* | *17.6* | *81.83* |
| | A-ViT† | 3.6 | 78.60 | | EViT‡ | 11.5 | 81.30 |
| | DynamicViT | 2.9 | 79.32 | | DynamicViT | 11.2 | 81.30 |
| | Evo-ViT‡ | 3.0 | 79.40 | | ToMe‡ | 11.5 | 81.41 |
| | ToMe | 2.9 | 79.49 | | DiffRate | 11.5 | 81.76 |
| | EViT‡ | 3.0 | 79.50 | | METR+DiffRate | 11.5 | 82.05 |
| | METR+EViT‡ | 3.0 | 79.76 | | EViT† | 8.7 | 80.00 |
| | EViT | 3.0 | 79.32 | | DiffRate | 8.7 | 80.51 |
| | METR + EViT | 3.0 | 79.64 | | METR+DiffRate | 8.7 | 81.11 |
| | METR + EViT† | 3.0 | 80.09 | ViT-B (MAE) | *Baseline* | *17.6* | *83.72* |
| | DiffRate | 2.9 | 79.72 | | ToMe‡ | 11.5 | 82.94 |
| | METR + DiffRate | 2.9 | 79.83 | | DiffRate | 11.5 | 83.25 |
| | DynamicViT | 2.5 | 78.50 | | METR+DiffRate | 11.5 | 83.29 |
| | EViT‡ | 2.3 | 78.50 | | ToMe‡ | 8.7 | 81.91 |
| | METR+EViT‡ | 2.3 | 79.06 | | DiffRate | 8.7 | 81.97 |
| | EViT | 2.3 | 78.47 | | METR+DiffRate | 8.7 | 82.42 |
| | METR + EViT | 2.3 | 78.94 | ViT-L (MAE) | *Baseline* | *61.6* | *85.95* |
| | METR + EViT† | 2.3 | 79.62 | | ToMe‡ | 31.0 | 85.05 |
| | DiffRate | 2.3 | 79.22 | | DiffRate | 31.0 | 85.15 |
| | METR + DiffRate | 2.3 | 79.46 | | METR+DiffRate | 31.0 | 85.32 |

### 4.3 COMPARE WITH EXISTING METHODS

In this section, we compare with state-of-the-art methods to demonstrate the effectiveness of METR. We select EViT (Liang et al., 2021) and DiffRate (Chen et al., 2023b) as the base method, and build METR on top of them. Note that for these two base methods, we report our re-produced results. Additionally, DynamicViT (Rao et al., 2021), A-ViT (Yin et al., 2022), Evo-ViT (Xu et al., 2022), and ToMe (Bolya et al., 2022) are compared. For holistic evaluation, we consider both the from-scratch training setting and the fine-tuning setting. The results are shown in Tab 5.

Again, we observe stable improvement brought by METR. Specifically, the improvement is robust to different base methods, different models, and different schedules. Such enhancement convincingly proves the existence and the effectiveness of the special chemistry between multi-exit architecture and token reduction. Note that since METR investigates transformer token reduction from an aspect different from most existing works on this topic, it can boost state-of-the-art to a new level. For example, even though the recent DiffRate (Chen et al., 2023b) has already surpassed other methods by a large margin, METR can still enhance it in a stable and considerable manner. On the other hand, there is still a clear tendency that *more aggressive token pruning benefits more from METR*. For example, considering the ViT-B (DeiT) model, METR improves DiffRate by 0.29% in the GFLOPs=11.5 setting, and the margin increases to 0.6% with GFLOPs=8.7.

## 5 CONCLUSION

In this paper, we scrutinize the congruence between [CLS] attention and token significance, a foundational premise underlying existing token reduction approaches, and pinpoint its intrinsic unreliability. To ameliorate this issue, we introduce METR—a straightforward yet efficacious amalgamation of multi-exit architecture and token reduction. Extensive empirical evaluations corroborate that the early task pressure, engendered by the multi-exit loss, is highly effective in calibrating [CLS] attention. Furthermore, this adjustment can elevate existing token reduction methods to a new level.

## 6 ACKNOWLEDGEMENT

This work was partially supported by the Natural Science Foundation of China (Nos. U2336213 and 62122074) and Innovation Funding of ICT, CAS under Grant No.E000000.

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

## A  Experiment details

The experiment hyper-parameters are mostly inherited from the base method on which METR is built. All experiments are conducted on a server with 8 NVIDIA RTX 4090 GPUs.

### A.1  METR + EViT

Under the 30-epoch fine-tuning setting, the official DeiT (Touvron et al., 2021) checkpoints are loaded as model initialization. The model is then trained with effective batch size 2048, actual learning rate (after proportional scaling w.r.t. effective batch size) 8e-5, minimum learning 2e-6, without learning rate warm up. The cosine learning rate scheduler is adopted. The optimizer is AdamW, with weight decay set to 1e-6. For self-distillation, the temperature for distillation loss (Hinton et al., 2015) is set to 1, and the weight, namely $\alpha$ in Eq. 10, is set to 1.0; when $\mathcal{L}_{me}$ instead of $\mathcal{L}_{sd}$ is used for ablation studies, the weight for $\mathcal{L}_{me}$ is also 1.0, as we find that $\mathcal{L}_{me}$ and $\mathcal{L}_{sd}$ generally share the same optimal weight.

For the 100-epoch finetuning setting, the learning rate is enlarged to 8e-4, and the other hyper-parameters are left unchanged.

For the from-scratch training setting, the model is trained for 300 epochs. Consistent to the original setting of DeiT, the learning rate and minimum learning rate are set to 2e-3 and 1e-5, the number of warm-up epochs is set to 5, and weight decay is set to 0.05. The training starts with token keep ratio equal to 1, namely no token reduction is applied; subsequently, from epoch 100 to epoch 150, the keep ratio linearly decays to the target value; finally, after epoch 150, the keep ratio is fixed to the target value. Different from the fine-tuning settings, we find that smaller $\alpha$ (weight for $\mathcal{L}_{sd}$), like 0.3, makes the best results, while larger $\alpha$ will hurt the performance.

For all of the above settings, token reduction is applied in the 4-$th$, 7-$th$, and 10-$th$ blocks, with inattentive token fusion (proposed in the EViT (Liang et al., 2021) paper) turned on. Correspondingly, the early-exit heads are placed in these blocks.

### A.2  METR + DiffRate

For experiments based on DiffRate (Chen et al., 2023), we adopt the Prune & Merge token reduction operation, and the searched token reduction schedule (i.e. how may tokens should be reduced in each block). Unlike EViT, DiffRate densely applies token reduction in every block; as supported by Tab. 4, we correspondingly add early-exit heads to all these blocks except the first and the last one.

All METR + DiffRate experiments follow the 30-epoch finetuning setting, where the officially released checkpoints for DeiT (Touvron et al., 2021) and MAE (He et al., 2022) (after supervised finetuning) are used for initialization. For all experiments, we set effective batch size 1024, actual learning rate (after proportional scaling w.r.t. effective batch size) 8e-5, minimum learning 1e-6, 1 epoch of learning rate warm up, and weight decay 0.05. As an exception, for the ViT-L (MAE) model, the learning rate is set to 2e-5. The weight $\alpha$ for $\mathcal{L}_{sd}$ is set to 1 accross all experiments.

## B  Discussion: off-the-shelf v.s. token-reduction aware finetuning

Comparing Tab. 1 with other experiments in the main text, we find that the advantage of METR is numerically more salient in the off-the-shelf setting. We deduce that the reason is that, with finetuning, the model can make some compromises, e.g. redundantly copy and store information among tokens, so that it could gain some robustness even when the token scoring metric is not reliable. In contrast, the off-the-shelf setting is way more challenging and poses stricter demands on the accuracy of token scoring. To empirically validate this deduction, we consider the random token selection baseline. Using the official DeiT-S checkpoint (79.80 top-1 accuracy), we first directly evaluate the model in the off-the-shelf setting, with token reduction applied and keep ratio set to 0.5. When [CLS] attention is used for token scoring and selection, the accuracy drops to 74.10; in contrast, when tokens are random reduced, the accuracy drops dramatically to 71.46. We then consider the 30-epoch finetuning setting, employing $\mathcal{L}_{final}$ as loss function, with token-reduction

operation applied. When [CLS] attention is used for finetuning and evaluation, the accuracy is 78.47; when random token selection is used for finetuning and evaluation, the accuracy is 77.12 – the gap is much smaller than the previous setting. This contrast provides an explanation over why the advantage of METR is more pronounced under the off-the-shelf setting.

However, as indicated by our experiment results in main text, even though the improvement in the finetuning setting is not as numerically salient as the off-the-shelf setting, it is still fairly considerable, especially when taking the average developing pace of the token reduction topic into consideration. It indicates that an accurate token scoring metric is still indispensable in fulfilling the potential of token reduction, even with token-reduction aware finetuning applied. An explanation is that, the aforementioned compromises are unlikely to *completely* erase the influence of scoring metric quality. Furthermore, the more unreliable the metric is, the stronger the comprises have to be and the more model capacity would be occupied to achieve it, which would inevitably hurt the representation ability and the accuracy of the model.

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
