# OpenReview forum: "A Simple Romance Between Multi-Exit Vision Transformer and Token Reduction"
_ICLR.cc/2024/Conference — ICLR 2024 poster_

### Official Review · Reviewer_G39w · 2023-10-28

**Soundness:** 3 good
**Presentation:** 3 good
**Contribution:** 3 good
**Rating:** 8
**Confidence:** 4

**Summary:**

This paper combines two well-established methods, multi-exit neural networks and vision transformer (ViT) token reduction, to improve the efficiency of ViT. The background is that the prominent ViT token reduction techniques like EViT are based on removing unimportant tokens based on the attention scores that naturally indicate the contribution of the visual tokens to the final ViT prediction. The authors' motivation is that in the previous method, the ViT had no incentive to make sure the attention scores in the shallow layers aligned with the semantic importance of the visual tokens. To motivate the ViT to have such an incentive, the authors propose to exploit the multi-exit training method with ViT exits in the intermediate layers, which requires early class information fusion via the attention scores, thus ensuring the attention scores exhibit semantic importance properties. With the combination of the two techniques, the authors show a noticeable improvement compared to the baseline, especially when a significant number of tokens are removed or with more finetuning epochs.

**Strengths:**

- The method is clearly motivated. Token reduction based on the existing attention scores in ViT has been shown to be an effective method in reducing computational costs while maintaining most of the classification accuracy. The authors propose to add the pressure of extracting the classification information in the shallow layers via the multi-exit mechanism, which forces the attention scores in the shallow layers to focus on the important tokens as the scores are directly used as weights to gather the information for classification in the multi-exits.
- The experiments are extensive and show the effectiveness in improving the classification accuracy over the baselines, especially with a longer training schedule. Experiments also demonstrate the proposed method's effectiveness from different perspectives, including different base models (DeiT/MAE), different model sizes, and different base methods (EViT/DiffRate).
- The visualization seems to support the claim that adding multi-exit modules to the ViT makes the attention scores in the shallow layers aligned better with human perception (higher scores are allocated to more important tokens on the objects).

**Weaknesses:**

The weaknesses are mostly minor issues, but it is important to address them to make the paper clearer and easier to understand.
- The phrase "Reduce Ratio" is not a good term to indicate the ratio of how many tokens are kept. Please change to another term like "keep ratio" to make it clear.
- Table 1 is not well explained. It took me a while to understand the setting of the experiment. The term "Off-the-shelf" is not immediately understandable. It would improve clarity by explicitly explaining the details of the experiments.
- It would improve the readability of the paper by changing some words/notations to standard ones, e.g., CSE -> CE for cross-entropy.
- There seems no appendix, but at the end of Section 4 first paragraph, it says "See appendix for detailed experiment settings."
Please carefully proofread the whole paper to address these nuanced issues.

**Questions:**

- Is the loss $L_{me}$ in Eq (8) also added to the $L_{total}$? It is not clearly mentioned in the paper.
- Figure 2 can be improved with better illustration and explanation. Specifically, the arrows from the $A^c$ to the [CLS] are somewhat confusing. And why do the patch tokens have fading colors from the bottom up?

---

> ### Author Response · Authors · 2023-11-16
>
> Thanks for your recognition of our work and the constructive suggestions!
>
> ### Enhancing readability
>
> Based on your comments, we have made the following modifications in the revision:
>
> + We have changed the term from "reduce ratio" to "keep ratio" to refer to the proportion of tokens to keep. We have also added an elaboration of this concept in the main text.
> + We have refined the introduction of the off-the-shelf setting in both the main text and the caption of Tab. 1. It should now be clearer.
> + We have changed the notation for cross entropy to $\mathrm{CE}$.
>
> ### Appendix
>
> Sorry for the absence of the appendix in our initial submission. It is now available in the revision.
>
> ###  $\mathcal{L}\_{me}$ and $\mathcal{L}\_{total}$
>
> $\mathcal{L}\_{total}$ does not contain $\mathcal{L}\_{me}$. Specifically, when self-distillation is applied, $\mathcal{L}\_{sd}$ will directly replace $\mathcal{L}\_{me}$, thus they do not coexist.
>
> ### About Fig. 2
>
> Sorry for making you confused. By drawing the blue arrows from $A^{c}$ to the $[\textrm{CLS}]$ token, we hope to convey the idea that information from other tokens flows into the $[\textrm{CLS}]$ token, with $A^{c}$ defining the contribution ratio of each token. In the revision, we have added the explanation to the caption of the figure. The fading colors are used merely to differentiate the patch tokens, enabling the demonstration of the fact that the order of the tokens is changed after token sorting. Based on your comment, we have canceled the fading colors and instead labeled the tokens with id numbers.

---

### Official Review · Reviewer_jWbQ · 2023-10-31

**Soundness:** 3 good
**Presentation:** 3 good
**Contribution:** 3 good
**Rating:** 6
**Confidence:** 2

**Summary:**

The paper introduces METR, a straightforward approach that combines multi-exit architecture and token reduction to decrease the computational burden of vision transformers (ViTs) while maintaining accuracy. The authors discover a discrepancy between the attention score of [CLS] and the actual importance of tokens in early ViT blocks, which negatively affects the performance of token reduction methods relying on this metric. The authors demonstrate that METR can improve existing token reduction techniques and achieve better results than state-of-the-art methods on standard benchmarks, particularly when using high reduction ratios.

**Strengths:**

Overall, METR is a promising method that can help reduce the computational cost of ViTs while maintaining accuracy.

- The paper is clear and well-motivated.
- The idea is intriguing and demonstrates significant improvement compared to other baselines.
- The evaluation is well-designed and highlights the core contribution in the design section.

**Weaknesses:**

- The evaluation demonstrates a notable improvement in accuracy compared to the baseline frameworks. It will be helpful to further demonstrate the reduction in latency with fewer FLOPs compared to other baselines.
- It would be beneficial if the author could offer more insights in the method section, such as explaining how and why this design can enhance performance.

**Questions:**

Please see above

---

> ### Author Response · Authors · 2023-11-16
>
> Thank you for your comments!
>
> ### Latency
>
> Given that we have already reported the FLOPs in the initial submission, I guess that by mentioning latency, you mean that we should further report the wall-clock latency or actual throughput to demonstrate the effect of METR in practice (If we misunderstood, please correct us).  We would like to first clarify that such indicators were absent in the initial submission because the latency and throughput are METR is totally determined by the base method with which METR is combined (e.g. EViT and DiffRate), rather than METR itself, as METR changes neither the architecture nor the computation at inference time. In other words, while METR can improve the accuracy of existing methods under the same latency, the exact latency is not defined or influenced by METR.  Therefore, we respectfully believe that, unlike most existing works in the token reduction area, latency and throughput are not of great importance in judging the effectiveness of METR. However, we also agree that listing such indicator directly in our paper will make it easier for the readers to form an intuitive understanding over the effect of METR in practice. Therefore, we have supplemented some throughput data in the revision.
>
> ### Adding more insights in the method section
>
> We deeply appreciate your suggestions. The fundamental insight and motivation behind METR are primarily detailed in Section 1. Additionally, we have succinctly reiterated the motivation in Section 3.2. As we aim to clarify any ambiguities and enhance our presentation, could you kindly provide more specific details regarding any aspects that may currently seem unclear? Your feedback would be invaluable in guiding our improvements.

---

### Official Review · Reviewer_Zdao · 2023-10-31

**Soundness:** 3 good
**Presentation:** 3 good
**Contribution:** 3 good
**Rating:** 8
**Confidence:** 4

**Summary:**

This submission introduces METR, a simple and effective technique for informed token reduction applied in Vision Transformer-based image classification. An analysis presented in the manuscript, demonstrates that the commonly used [CLS] token attention scores, acting as an importance metric for token pruning, are far more effective on deeper blocks in contrast to shallower ones. This is attributed to the long gradient distance from the task loss, traditionally applied at the end of the network.

To remedy this, the manuscript proposes the introduction of intermediate classifiers at training time, forming a multi-exit transformer model, in which all token reduction blocks are exposed to stronger task supervision. Upon deployment, early-exits are removed eliminating any speed overhead, while extensive experiments demonstrate the effectiveness of the proposed method across different models and in comparison with several baselines.

**Strengths:**

-The manuscript focuses on the very interesting interplay between multi-exit models and token reduction.

-Sec. 3.2, introduces a simple, yet effective solution to the examined problem. The discussion on the use of attention as an importance metric is insightful and many relevant works can benefit from these findings.

-Experiments are extensive in terms of examined models and baselines, and validate the superiority of the proposed approach to the baselines.

-The manuscript is generally well-written and easy to follow.

**Weaknesses:**

-The use of self-distillation loss for the multi-exit training (Eq.9), in place of traditional multi-exit loss of Eq.8, although effective, is not adequately motivated. Self-distillation is typically used to improve the accuracy of the trained exits, which is not a requirement here as these are discarded at inference time. The manuscript would benefit from a more insightful analysis of what motivated this design choice/ why do the authors believe this works better than the traditional approach.

-Row(2) in Tab.3 seems to be the equivalent of row (4) in Tab.2, where multi-exit and token-reduction fine-tuning are jointly applied (instead of the two-stage ablation in Tab2). If this is the case, it can be deduced that token-aware fine-tuning notably reduces the effectiveness of the proposed approach, leading to significantly smaller gains even when aggressive token reduction takes place. This fact is separate from the commented fading of multi-exit effects after separate fine-tuning and needs to be further investigated/discussed in the manuscript.

Note: An appendix is mentioned in the manuscript (Sec.4), but was not accessible to the reviewer.

**Questions:**

1. What motivated the use of self-distillation in place of traditional multi-exit training in the proposed setting? What are the authors' insights about the demonstrated effectiveness of this design choice?

2. Is token-reduction aware fine-tuning indeed limiting the effectiveness of the proposed approach? If yes, this should be commented in the manuscript.

3. In Tab.1,2,3 does “reduce ratio” refer to number of tokens or GFLOPs? Both should be reported to get the full picture.

Minor comments/ Presentation:
-Notation in Sec.3 is quite loose. Consider defining the dimensionality of each introduced symbol (X,x,A,...).
-Sec3.2: Symbol a^{c-p} is confusing.
-Sec4.1.1: without incorporate -> without incorporating,  Subsequently, We (...) -> Subsequently, we (...)

---

> ### Author Response · Authors · 2023-11-16
>
> Thank you for your recognition and constructive suggestions!
>
> ## About Self-Distillation
>
> While the working mechanism of knowledge distillation is not yet fully revealed, recent studies have made some important steps in identifying the aspects from which knowledge distillation may benefit. Among them, the following points could help explain why self-distillation would benefit METR:
>
> 1. Label denoising  [1]: existing datasets, though already curated, more or less suffer from inaccurate labeling. Aside from the images whose labels are literally wrong due to mistakes and errors in the data collection process, problems like the co-existence of multiple classes also lead to noises, as the label is always one-hot. Such data noise is adverse to generalization. When it comes to the early-exit heads of METR, this will also mislead the $[\mathrm{CLS}]$ token in determining which patch tokens to attend to. Self-distillation can alleviate this problem to some extent, as the soft predictions from the teacher model (the full-depth model in our case)  make better approximation to the real class distributions of the images.
> 2. Another possible advantage of self-distillation is sample reweighting [2]. The inherent property of KL-Divergence causes the fact that the gradient applied to the student is magnified when the teacher model is confident over the sample; when the teacher is uncertain about a sample, the magnitude of student gradient is weakened.  This again plays the role of dataset purification, as the teacher model is more likely to be uncertain about problematic samples than unambiguous ones. When trained with such cleaner supervision, it would be easier for $[\mathrm{CLS}]$ token to learn to differentiate between important and unimportant patch tokens.
> 3. Additional cues [3]. As pointed out in the original KD paper, probability distribution predicted by the teacher on non-target classes provides additional cues, which could be helpful for the student. The first point listed above (label denoising) can be regarded as the embodiment of this idea on inaccurately-labled samples. On the other hand, even on accurate samples, such implicit cues might also provide some advantage. For example, given two images of dog, suppose that from the appearance perspective, the first one looks more similar to a fox, while the second one looks more similar to a wolf. The soft labels predicted by the final classification head may then reflect such similarity, i.e. allocate more probability to class fox for the first image, and more probability to class wolf for the second image. With self-distillation, such signals will serve as additional cues, which not only enhance the accuracy of early heads, but more importantly motivate the $[\mathrm{CLS}]$ token to allocate its attention in a way through which the $[\mathrm{CLS}]$ token can produce similar representations. For example, for the first image, the $[\mathrm{CLS}]$ token would be driven to pay attention to contents like ears, which grant the dog with fox-like looking. This may then benefit token reduction, as the tokens with high $[\mathrm{CLS}]$ attention scores tend to contain richer information and details.
>
> [1] Feature normalized knowledge distillation for image classification. ECCV, 2020.
>
> [2] Understanding and Improving Knowledge Distillation. arXiv, 2020.
>
> [3] Distilling the knowledge in a neural network. NIPS Deep Learning Workshop, 2014.

---

> ### Author Response · Authors · 2023-11-16
>
> ### Does fine-tuning hurt the effectiveness of METR?
>
> Following your suggestion, we have added a discussion about this problem in Appendix B. For your convenience, we copy the contents here:
>
> > Comparing Tab. 1 with other experiments in the main text, we find that the advantage of METR is numerically more salient in the off-the-shelf setting. We deduce that the reason is that, with finetuning, the model can make some compromises, e.g. redundantly copy and store information among tokens, so that it could gain some robustness even when the token scoring metric is not reliable. In contrast, the off-the-shelf setting is way more challenging and poses stricter demands on the accuracy of token scoring. To empirically validate this deduction, we consider the random token selection baseline. Using the official DeiT-S checkpoint (79.80 top-1 accuracy), we first directly evaluate the model in the off-the-shelf setting, with token reduction applied and keep ratio set to 0.5. When $[\mathrm{CLS}]$ attention is used for token scoring and selection, the accuracy drops to 74.10; in contrast, when tokens are random reduced, the accuracy drops dramatically to 71.46. We then consider the 30-epoch finetuning setting, employing $\mathcal{L}_{final}$ as loss function, with token-reduction operation applied. When $[\mathrm{CLS}]$ attention is used for finetuning and evaluation, the accuracy is 78.47; when random token selection is used for finetuning and evaluation, the accuracy is 77.12 -- the gap is much smaller than the previous setting. This contrast provides an explanation over why the advantage of METR is more pronounced under the off-the-shelf setting.
> >
> > However, as indicated by our experiment results in main text, even though the improvement in the finetuning setting is not as numerically salient as the off-the-shelf setting, it is still fairly considerable, especially when taking the average developing pace of the token reduction topic into consideration. It indicates that an accurate token scoring metric is still indispensable in fulfilling the potential of token reduction, even with token-reduction aware finetuning applied. An explanation is that, the aforementioned compromises are unlikely to *completely* erase the influence of scoring metric quality. Furthermore, the more unreliable the metric is, the stronger the comprises have to be and the more model capacity would be occupied to achieve it, which would inevitably hurt the representation ability and the accuracy of the model.
>
> ### Typos and notations
>
> Thank you for pointing out these problems! We have corrected the typos, and refined the mathematical notations based on your suggestions.
>
> ### About 'Reduce Ratio'
>
> In the initial submission, we used the term 'Reduce Ratio' to refer to the ratio of tokens to $keep$ at each token-reduction block. As pointed out by you and Reviewer G39w, we agree that the phrase is not proper here, and we have changed it to 'Keep Ratio' in the revision. Furthermore, we have also added an elaboration of this concept in the main text.
>
> ### Appendix
>
> Sorry for the absence of the appendix in our initial submission. It is now available in the revision.

---

> ### Comment · Reviewer_Zdao · 2023-11-20
>
> Re: Distillation - Thanks for this insightful analysis. I believe the above explanation of the intuition behind the design choice of employing the Self-Distillation loss actually adds value to the paper. As such, I would encourage you to try and "distil" the most important point of the above in one sentence (or so), and add it as motivation in Sec 3.3 and/or 4.2.2 of the manuscript.
>
> Re: Finetuning - The added discussion adequately covers my concern. I believe an additional ablation experiment that employs similar fine-tuning but including the multi-exit loss will better indicate what percentage of the accuracy gap is created by the fading of multi-exiting, vs the drift caused by finetuning (in case you find the time to include this in the appendix of camera ready version).
>
> Following the above, I am happy to increase my score to Accept.

---

> > ### Author Response · Authors · 2023-11-22
> >
> > Dear Reviewer Zdao,
> >
> > Thank you very much for recognizing our previous response and for the increased score! We appreciate your valuable feedback and agree with your following suggestions. We will make the necessary refinements to our manuscript in accordance with your comments for the final version.
> >
> > Sincerely,
> > Paper 1780 Authors

---

### Official Review · Reviewer_9w9u · 2023-11-02

**Soundness:** 3 good
**Presentation:** 3 good
**Contribution:** 3 good
**Rating:** 8
**Confidence:** 5

**Summary:**

This work has proposed a new token-pruning method, by integrating the multi-exit strategy into ViT. This work diagnoses the inconsistency between [CLS] attention and token importance in early ViT block, which degrades the performance of token reduction methods. To tackle this problem, this work introduces multi-exit architecture that allows the [CLS] token to gather information pertinent to the task in the early blocks. It also adopts self-distillation to improve the quality of early supervision. As a results, it achieves state-of-the-art performance.

**Strengths:**

### Good Motivation
This work has adeptly identified and proposed solutions for a problem in the literature of token pruning method of ViT.

### Novelty and SOTA performance
To address the inconsistency between [CLS] attention and token significance at the early blocks, the proposed method that incorporates multi-exit into ViT) is novel and it shows effectiveness clearly by achieving state-of-the-art performance.

### Nice visualization
This work shows well-supportive visualization examples.

**Weaknesses:**

No exists.

**Questions:**

It would be better to shows the GPU-throughput and compare it with those of SOTA.

---

> ### Author Response · Authors · 2023-11-16
>
> Thank you so much for your recognition of our work!
> ## About GPU throughput
>
> We would like to first clarify that we did not report throughput in the initial submission because the latency and throughput of METR are *totally* determined by the base method with which METR is combined (e.g. EViT and DiffRate), rather than METR itself, as METR changes neither the architecture nor the computation at inference time. In other words, while METR can improve the accuracy of existing methods under the same throughput, the exact throughput is not defined or influenced by METR at all. Therefore, we respectfully believe that, unlike most existing works in the token reduction area, latency and throughput are not of great importance in judging the effectiveness of METR. However, we also agree that listing such indicator directly in our paper will make it easier for the readers to form an intuitive understanding over the effect of METR in practice. Therefore, we have supplemented some throughput data in the revision.

---

### Meta-Review · Area_Chair_7B1Y · 2023-12-07

**Metareview:**

Multi-Exit Token Reduction (METR) improves the computational efficiency of Vision Tranformers (ViTs) by skipping computation for certain tokens, known as token reduction, according to scores from intermediate classifiers across the depths of the architecture, known as multiple exits. While there are many recent methods for token reduction, and there are a variety of multi-exit architectures, this work highlights and demonstrates the particular value of their combination by evaluation against a battery of current baselines and analyses by ablations. The experiments measure computation in terms of keep/skip ratio, FLOPs, and wallclock throughput (images/s) and measure model accuracy on the gold standard vision recognition benchmark of ImageNet. The model ablations support that the full method achieves the best reduction in computation and the highest accuracy. The comparison with existing methods also shows small but consistent improvements in accuracy for matched computational costs, and the accuracy gap grows with more aggressive token reduction (as seen by comparing accuracies across different GFLOPs limits). All reviewers side with acceptance and agree on the state-of-the-art accuracy/efficiency of METR, the justified motivation for the work, and its clear exposition (especially following a few minor fixes in the revision).

As a point of miscellaneous feedback, the authors are encouraged to do a last pass for proofreading ("chesmistry", "in fact, [...] that this is really the fact", ...) and terminology ("task pressure" vs. "loss" or a more common term).

**Justification For Why Not Higher Score:**

The experiments have sufficient scope across architectures, base methods, and computational budgets to show small but consistent improvements for image classification. However, there are no results on other datasets or tasks, event though ViTs are now common across recognition tasks like detection and segmentation. In fact the abstract says "METR outperforms state-of-the-art token reduction methods on standard benchmarks" but there is a single benchmark (ImageNet). Furthermore, the gains are for less than 1 point absolute in accuracy in all cases. Lastly, while this work is technically and empirically sound and thorough, the two ingredients in its "chemistry" of multi-exit architecture and token reduction are well-known.

**Justification For Why Not Lower Score:**

The work is well-reasoned and justified (see Fig. 1 and Sec. 1), well-executed technically and explained in detail (see overview in Sec. 3.1 and Fig. 2 along with the experimental detail of Sec. 4), and and the results are adequate. Rejection would deny that the work is correct, informative, and an improvement on the a popular topic of how to improve the efficiency of ViTs.

---

### Decision · Program_Chairs · 2024-01-16

Accept (poster)